# Energy Prediction and Optimization for Smart Homes with Weather Metric-Weight Coefficients

**DOI:** 10.3390/s23073640

**Published:** 2023-03-31

**Authors:** Asif Mehmood, Kyu-Tae Lee, Do-Hyeun Kim

**Affiliations:** 1Smart Information Technology Engineering Department, Kongju National University, Cheonan 31080, Republic of Korea; asif@kongju.ac.kr; 2Computer Engineering Department, Jeju National University, Jeju 63243, Republic of Korea

**Keywords:** smart home, appliance energy, energy prediction, energy optimization

## Abstract

Home appliances are considered to account for a large portion of smart homes’ energy consumption. This is due to the abundant use of IoT devices. Various home appliances, such as heaters, dishwashers, and vacuum cleaners, are used every day. It is thought that proper control of these home appliances can reduce significant amounts of energy use. For this purpose, optimization techniques focusing mainly on energy reduction are used. Current optimization techniques somewhat reduce energy use but overlook user convenience, which was the main goal of introducing home appliances. Therefore, there is a need for an optimization method that effectively addresses the trade-off between energy saving and user convenience. Current optimization techniques should include weather metrics other than temperature and humidity to effectively optimize the energy cost of controlling the desired indoor setting of a smart home for the user. This research work involves an optimization technique that addresses the trade-off between energy saving and user convenience, including the use of air pressure, dew point, and wind speed. To test the optimization, a hybrid approach utilizing GWO and PSO was modeled. This work involved enabling proactive energy optimization using appliance energy prediction. An LSTM model was designed to test the appliances’ energy predictions. Through predictions and optimized control, smart home appliances could be proactively and effectively controlled. First, we evaluated the RMSE score of the predictive model and found that the proposed model results in low RMSE values. Second, we conducted several simulations and found the proposed optimization results to provide energy cost savings used in appliance control to regulate the desired indoor setting of the smart home. Energy cost reduction goals using the optimization strategies were evaluated for seasonal and monthly patterns of data for result verification. Hence, the proposed work is considered a better candidate solution for proactively optimizing the energy of smart homes.

## 1. Introduction

Home appliances are considered to account for a large portion of smart home energy usage because appliances have increased in number in recent years. This is due to the introduction of many new home appliances that help consumers enjoy a comfortable lifestyle [1]. In fact, this comfort comes with the costs associated with home appliances used to support modern and ever-evolving technologies. Some of these IoT devices [2] are dishwashers, refrigerators, microwaves, and smart cars [3].

Due to the increase in appliances, energy efficiency has become a major challenge for many organizations. With an ever-growing population, increasing demand for energy, and the need to reduce energy consumption in smart homes, it is increasingly important to find ways to make energy consumption more efficient and sustainable. However, this can be challenging due to the complexity of the processes and systems involved. Using data to predict energy efficiency can be an effective solution to this problem. Data can be used to address many different types of inefficiencies.

The IoT [4] in the energy field is rapidly expanding as it becomes more affordable and efficient. Recently, some startups have focused on developing small home-appliance monitors to assist users at home. These devices can monitor weather conditions, appliance conditions, and other environmental information and send the data to a monitoring server. This technology enables energy systems to monitor smart homes without having to make house calls. In the future, IoT devices will enable users to deliver the remote control of energy settings around the world. This will allow us to create an eco-friendly environment.

CPS systems [5] are based on prediction, analysis, optimization, control, and scheduling, and are considered to be the solution to this domain [6]. An exemplary CPS architecture shown in Figure 1 is needed because of the diversity and variation involved in area and user-dependent energy data. In such systems, many smart home appliances could be monitored remotely using technologies such as wearables and smart-home devices. In addition, if intelligence is embedded into these movable devices/appliances [7,8], it is possible to enhance the performances of energy systems in many ways, e.g., energy optimization based on real-time monitoring. This could also help cut costs in smart homes by reducing the requirement for the user to stay at home for controlling the home appliances.

The CPS problems are divided into layers, such that there is a clear relationship between each entity of a layer. This enables an open design where dependency is removed by following and employing a service-oriented architecture. The main components involved in an energy-optimization system that are meant to solve complex problems are the predictor, optimizer, scheduler, and controller, which can be visualized in Figure 1.

Monitoring [9] of smart home and healthcare appliances could be done through the use of technologies such as wearables and smart devices [10], as depicted in Figure 1. It enables the energy systems to get more insights into the data patterns, which could help in planning the future efficiently. This could also help cut costs in smart homes by reducing the requirement for the user to stay at home by controlling the home appliances, thereby reducing the other complications associated with the optimization of smart home energy control systems. It could also help maintain a healthy environment with fewer errors and streamline administrative tasks. In this work, we have used the appliance energy data [11,12] and the information related to energy-system considerations [13,14].

Optimization is an important part of saving resources [15]. Regarding energy, a variety of optimization methods can be performed by tweaking a few settings in the home, office, or hospital [16]. Some of these settings are manually adjusted, such as controlling temperature and humidity by carefully switching appliances on and off at the correct times of the day. For this purpose, optimization techniques are used in machine learning techniques to train the model.

Current optimization techniques somewhat reduce energy usage but overlook user convenience, which is the main goal of introducing home appliances. Therefore, there is a need for an optimization method that effectively addresses the trade-off between energy saving and user convenience. In current optimization techniques, the inclusion of weather metrics other than temperature and humidity is also needed to effectively optimize the energy cost of controlling user-desired room settings. This research involves work for an optimization technique that addresses the trade-off between energy saving and user convenience and includes air pressure, dew point, and wind speed. To test the optimization, a hybrid approach utilizing GWO [17] and PSO [18] was modeled for this purpose. In addition, this work involves using appliance energy prediction [19] to enable proactive energy optimization. The LSTM model was designed for energy prediction. Both prediction and optimized control allow the proactive and effective control of smart home appliances, and can be evaluated through the results provided.

The optimization layer shown in Figure 1 refers to finding an optimal solution to the problem under consideration. This can be related to minimizing or maximizing the target variable. For this, an objective function was designed, implemented, and evaluated, which helps in making the decision of selecting the best candidate algorithm for optimization. It allows the system to make decisions about system scheduling and control so that the problem is solved accordingly. This layer, i.e., optimization, uses many algorithmic techniques, such as PSO, GWO, GA, Bayesian optimization.

Prediction is a way of knowing the future. From Figure 1, provision of better control over energy systems can be realized in the domain of CPS. Making better use of data in this way can also help to improve the reliability and availability of the system, which can be an important benefit in environments where resources may be limited. Being able to predict [20] and prevent many problems ahead of time can make it easier to implement changes and improvements as necessary, saving time and increasing productivity overall.

Many companies are already using predictive data analytics to improve their energy efficiency and reduce costs. IBM is an example of a company that is applying this approach in many areas, including energy management systems, supply-chain optimization, and urban infrastructure management. IBM has worked with a wide range of clients, including large government agencies and private companies, such as Walmart and FedEx. The company has also developed a number of its products that use predictive data analytics to improve energy efficiency and reduce costs.

The prediction layer is responsible for providing relevant predictions about the environmental metrics involved in system. For this, machine learning techniques are used, e.g., LSTM [21,22,23]. For data patterns that change over time, RNN techniques are used. Similarly, many techniques can be applied according to data attributes.

Scheduling [24] the appliances is very important, as it may affect the energy consumption and device performance. Thus, it must be carefully planned. The information being used in decisions for control devices has turned out to be very handy, as it provides knowledge and the state of the environment. This includes both operational data and information about existing systems, and data about current patterns of energy use and information about future requirements. When used effectively, data can be used to predict and prevent many of these problems. This can help to improve efficiency and reduce operating costs. It can also help reduce the environmental impact of operations by improving the sustainability of the system and reducing the use of energy resources in an optimized way. For example, data collected during the testing of new equipment can be used to create a system model that can be used to accurately predict the performance of new equipment when it is installed into an existing system.

Scheduling and controlling [25,26] are the next steps after finding the best solution to the identified complex problem and can be visualized in Figure 1. These techniques give us the opportunity to apply the optimal appliance configurations and their settings to save the energy in smart homes. These steps allow the system to implement optimal solutions and control the environment, such that the systems; behavior can be analyzed before and after. To schedule tasks and control behavior, an IoT “app store” is used, in which tasks are mapped, assigned execution times, and deployed accordingly.

In addition, machine learning in smart applications [27] can be used to process and analyze vast amounts of data and improve energy utilization by taking control of the appliances and optimizing smart homes’ energy use. The advancement in technology has also made it possible to deploy AI-powered robots [28] in the operating room to perform control tasks while allowing the user to operate remotely using a control appliance attached to the robot. These movable devices can also assist with monitoring vital signs while functioning so that the time to change control parameters of the smart home can respond proactively to ensure the comfort of the user and optimization of energy at the same time.

This paper aims to provide a solution to the problem of maximizing the energy efficiency of a particular home through advanced machine learning algorithms. To attain this goal, we first discuss the various models that are used to build energy-optimization models for homes and the challenges that arise when implementing these models in the real world. We also introduce techniques that can be used to optimize energy usage based on machine learning. Thereafter, we compare the performances of these techniques and finally present our results. Following this, cost analyses are performed for identifying the monetary costs associated with achieving optimal energy efficiency from a particular home. Finally, we discuss these findings in the context of energy saving by providing energy management recommendations that will minimize energy costs and also improve the efficiency of these homes. This paper has a list of contributions to this domain of research.

Weather data analytics: We found the importance of each feature and estimated its impact on the target variable, i.e., appliance energy.Energy-optimization model: We modeled the energy-optimization model not only based on temperature and humidity but several other weather factors too, i.e., air pressure, dew point, and wind speed. We integrated their weight factors, which were calculated based on the importance of each feature.Energy-forecasting model: It was modeled to include the weight factors of each feature in LSTM. It was evaluated over different months, seasons, etc.

In Section 2, insights about existing literature are provided. Current mechanisms’ shortages are highlighted, which the proposal tends to solve by a proactive approach. In Section 3, details of the proposed system are provided, in which the prediction model and optimization algorithm are formulated. In Section 4, details about the proposed system’s implementation, experimentation, and evaluation are provided. The dataset, i.e., AEP, is also explained. Finally, the conclusive remarks are made and highlighted in Section 6.

## 2. Literature Review

Home appliances are thought to account for a large portion of the energy consumption of smart homes. This is due to the abundant use of IoT devices [29]. Various home appliances, such as heaters, dishwashers, and vacuum cleaners, are used every day. It is believed that a significant amount of energy use can be reduced through proper control of such home appliances. For this purpose, optimization [30] techniques focusing mainly on energy reduction are used. In addition, predictive techniques [21,22,23] are also used for the proactive control in smart homes. This section highlights the shortcomings of existing studies and then briefs the reader on the challenges.

For the optimization of energy, several works [30,31,32,33] have been proposed. In these works, different energy saving mechanisms and techniques have been described, proposed, and evaluated through the provision of justifiable results.

One author proposed a PSO-based optimization technique [31] such that it considers temperature and humidity metrics. This work also evaluates the proposed technique and tends to save a justifiable amount of energy. However, it does not consider other weather metrics such as air pressure, dew point, and wind speed. Considering these metrics is important because the cost of energy varies by season and refers to dependency on multiple factors, which could have an huge impact on the scope of optimization technique. Based on this understanding, we enhanced the optimization technique by utilizing a hybrid optimization technique, i.e., PSO-GWO.

Various optimization models [30,31] have been proposed, and the best model was proposed to improve the optimization performance. This includes the comfort factors in the optimization procedure. This involves the addition of user-desired temperature and humidity. By following this approach, the trade-off between user comfort and energy-cost saving could be made. However, this method also lacks the evaluation of optimization model without considering other weather metrics.

The authors from the articles [30,31] proposed a system consisting of an optimal technique to save energy of appliances in the smart-home environment. They utilized the PSO-based optimization technique to save in the energy use of appliances. This includes the use of two parameters for the overall house. However, there could be an enhancement in terms of bringing proactiveness to this process. This will not only enable before-hand optimal control of smart-home energy, but also enhance the monitoring systems of smart homes’ alarm notifications. The algorithms can be utilized by advanced energy systems [34,35] to manage the energy efficiently. Based on these studies, it has been found that the predictions enable proactiveness in controlling the room’s conditions according to the user-desired settings. The proposed solution enables this by providing energy optimization for the energy forecasts instead, which not only enables proactiveness, but allows the system to be prepared for alerting the consumer.

The authors of the article [32] proposed a system comprising PSO-based ensembled models for the energy forecast. This includes the feature-selection approach for enhancing the forecast accuracy. However, the analysis could be improved by extending it to different seasons, months, areas, weekends, weekdays, and times of the day. To this end, our proposal tends to include this evaluation in the energy forecasts by considering the weight factor of each feature, rather than just selecting the features. This evaluation was required to enhance the prediction model’s diversity. We performed a thorough examination of the data comprising 29 features, of which a few were selected based on the co-variance with appliance energy consumption. By doing so, we utilized these weight factors in the optimization model as well. The predictions considering the weight factors allowed us to improve the RMSE score.

Energy optimization can be achieved in many ways [36]. However, this paper focuses on analyzing the data and finding the importance of each feature for the optimization of energy use using machine learning techniques, and on minimizing the cost of energy consumption required for controlling the appliances. To achieve this, it is necessary to utilize certain machine learning techniques, such as PSO and GWO, that can be used for building the model to achieve optimal efficiency by altering the energy demand in a particular home. The model trained with these techniques was then validated using the data [11,12] in different cases to achieve the optimal level of energy conservation in a home. In addition, energy forecasting is also discussed to give an idea of the better control it provides, to save the energy in advance.

For the prediction of appliance energy, existing forecasting models [21,22,23] take into account the latest techniques. In these works, time-series forecasting methodologies, mechanisms, and techniques are described, proposed, and evaluated through the provision of justifiable results.

In [21], an LSTM ensemble network was trained to learn the adaptive weighting mechanism. Some techniques [22] utilize deep learning to improve the performance, whereas some of them [23] make short-term forecasts. However, the seasonal factor is missing in these works, which could be accounted for in the training process after the thorough analysis of seasonal data. Based on this analysis, they could be assigned weights varying over the season, month, weekdays, weekdays, etc. We tend to improve the accuracy of forecasting model [37] in the proposed system by evaluating the RMSE score of our model applied over the given datasets described in Section 4.2.

Overall, the proposed system tends to enhance the optimization for better control of energy use in smart homes. Firstly, for better control, a prediction model was designed. This was designed in such a way that it performs better in various weather conditions due to the consideration of weight factors of the selected features. This includes the addition of other weather metric weight factors for air pressure, dew point, and wind speed. Secondly, it enables proactiveness in the optimization problem of energy systems in smart homes. In addition, it includes evaluation factors that make the prediction/forecast model diverse in nature and applicable to many regions—the secondary objective of this work. Conclusively, according to the comparison of our results with the current literature [30,31,33], the other above-mentioned weather factors play important roles in saving energy better, as they also impact energy consumption due to the fact that each season’s energy consumption is different, thereby highlighting the requirement of such an optimization technique to cater to them. The proposed system’s results regarding prediction and optimization together prove the it outperforms the existing energy-optimization systems and enables better control over smart-home energy use.

## 3. Proposed System

This section is categorized into three sections, i.e., preprocessing, prediction, and optimization. Details on preprocessing the data and extraction of feature weight factors is described are Section 3.1. Description of appliance energy forecasting can be found in Section 3.2. The technique for controlling the appliances to save on energy in an optimal way is explained in Section 3.3.

### 3.1. Preprocessing

In this section, we describe the steps we followed to find the importance of features and to calculate the weights representing the impacts of features. This was required to enhance the performances of prediction and optimization models. In prediction, the weights were passed as an input during the training process, whereas the weights were applied while calculating the energy cost to control the appliances in the optimization simulations.

The preprocessing shown in Section 4.2 included the removal of nulls, zeros, and unnecessary features, and evaluating the importance of each feature against the target variable, i.e., appliance energy. These actions are required for enhancing the performance of the the proposed time-series forecasting model.

Table 1 describes the variables used for scaling the dataset.

Firstly, the unnecessary features must be removed—those which do not have any impact or may have redundant data. For this purpose, we utilized the covariance matrix values. Based on this analysis, some of the temperature and humidity features were removed from the dataset. The column lights were also removed due to abundance of zeros. All the filters were applied after thorough analysis, from the findings of correlation, zeros, nulls, etc.

Secondly, scaling dataset features is a required step [38] before training the model. This is because the existing models perform well over a smaller range of numbers. From the perspective of optimizers, the learning rate can easily be detected and enables a friendly environment for experimentation. Standard and min-max scalers are widely used for this purpose. Equation (Equation 1) depicts scaling of the dataset, a pre-processing step required before training. $X^{\prime }$ refers to the scaled dataset. It is retrieved after applying MinMax−Scalar to scale down data within a discrete set, i.e., between −1 and 1. The scalar value *x* refers to a value from a feature vector *X*; $x_{min}$ and $x_{max}$ refer to minimum and maximum scalar values from feature vector *X*, respectively. With the use of Equation (Equation 1), the dataset’s features are scaled as follows:(1)$$x^{\prime }=\frac{\left(x-x_{min}\right)\left(max-min\right)}{x_{max}-x_{min}}+min\;x\in X$$

Thirdly, weight factors of each feature are calculated. This is accomplished by applying the scaled data to multiple models for the selection of the best model configurations and training settings.

To get the best hyper-parameter [11] configurations for a model, we applied multiple models, e.g., MLP regressor, GBC, XTR, RF, SVR, Ridge, lasso, k-NN regressor, and XGB regressor. Among all these models, the extra tree regressor outputs the best results with the least RMSE score and maximum R2 score shown in Figure 2. Based on these results, we applied Boruta algorithm and GSCV to get the weight coefficients which refer to the importance of each feature represented by a scalar value. Weight coefficients can be retrieved by using the command grid_search.best_estimator_.feature_importances_. The importance values are then scaled within the range of 0 and 1, which are then used in the prediction and optimization problems.

The Boruta algorithm was used to identify the feature importance, i.e., weight coefficients. Based on all features’ importance values, the weight factor of each was calculated and assigned to each feature so that they could be applied in prediction and optimization mechanisms for better performance.

### 3.2. Prediction

Table 2 describes the variables used in the prediction problem.

The proposed appliance-energy-prediction model is shown Figure 3. It shows an LSTM based model which comprises input, hidden, and output layers, along with its LSTM units. It can be visualized that the temperature, humidity, air pressure, dew point, wind speed, etc., are passed in, along with their calculated weights. The addition of weight factors refers to the importance of that feature. This weight factor shows the impact of that feature over the target variable, i.e., appliance energy.

For the prediction, we utilize an LSTM layer, and then a fully connected dense layer. Applying features and their calculated weight factors enhances the performance of prediction over the temporal variation. The configuration of proposed LSTM model also involves setting the numbers of layers, neurons, epochs (50), and learning-rate (0.01) for better training. The data features used as an input for prediction of appliance energy are temperature, humidity, air pressure, dew point, and wind speed, along with their weight factors. Formulation of the proposed time-series prediction model based on LSTM is explained in the following section.

#### Problem Formulation

Formulation of prediction problem is required so that it is applied over the daily and hourly data for evaluation, such that the performance remains stable. The main procedures involved in LSTM prediction module are input gate $i_{t}$, forget gate $f_{t}$, and output gate $o_{t}$.

The LSTM model input gate is formulated in Equation (Equation 2), i.e., $i_{t}$. This gate decides what relevant data need to be added from the LSTM unit.
(2)$$i_{t}=\sigma \left(W_{xi}x_{t}+W_{hi}h_{t-1}+b_{i}\right)$$

The LSTM model forget gate is formulated in Equation (Equation 3), i.e., $f_{t}$. This gate decides which data need to be retained and which can be forgotten.
(3)$$f_{t}=\sigma \left(W_{xf}x_{t}+W_{hf}h_{t-1}+b_{f}\right)$$

The LSTM model output gate is formulated in Equation (Equation 4), i.e., $o_{t}$. This gate finalizes the next hidden state.
(4)$$o_{t}=\sigma \left(W_{xo}x_{t}+W_{ho}h_{t-1}+b_{o}\right)$$

The LSTM model regulator is formulated in Equation (Equation 5), i.e., $\tilde{c_{t}}$. It is used to regulate the vector values between −1 and 1.
(5)$$\tilde{c_{t}}=\phi \left(W_{xc}x_{t}+W_{hc}h_{t-1}+b_{c}\right)$$

LSTM model memory cell is formulated in Equation (Equation 6), i.e., $c_{t}$, which is retained and transferred to the next LSTM unit.
(6)$$c_{t}=\left(f_{t}\times c_{t-1}\right)+\left(i_{t}\times {\tilde{c}}_{t}\right)$$

The LSTM model unit’s hidden state is formulated in Equation (Equation 7), i.e., $h_{t}$. It decides to whether hidden state will be used in next LSTM unit or not. It is denoted as follows.
(7)$$h_{t}=o_{t}\times \phi \left(c_{t}\right)$$

Collectively, all Equations (Equation 2)–(Equation 7) mentioned above provide a network of LSTM units and their inter-connectivity. The configuration of LSTM also involves setting the number of layers, neurons, epochs, and learning rate for better training. The proposed time-series prediction model, i.e., LSTM, is modeled below.

### 3.3. Optimization

Table 3 describes the variables used in the optimization problem.

In this section, the optimization model is described, which includes the explanation of its internals and the formulation. In Figure 4, it can be seen that the proposed optimization makes the use of PSO-GWO hybrid approach and involves the other weather metrics, which are temperature, humidity, air pressure, dew point, and wind speed. In contrast, the previous approach only makes the use of temperature and humidity. Along with this, the proposed approach also makes the use of weight factors calculated through feature importance, as described in Section 3.1.

An optimization model is defined for the optimal control of the energy use of smart homes. The dataset [11,39] contains 29 features, from which only the most important features were selected, and the following formulation is be applied.

#### Problem Formulation

The optimization is formulated as per Equation (Equation 8). All dataset features are applied to the optimization formula to optimally reduce appliance energy consumption by considering the weights associated with each feature based on their coefficients.
(8)$$\vec{X}\left(t+1\right)=\frac{\vec{\alpha }+\vec{\beta }+\vec{\gamma }}{3}$$

Equation (Equation 9) refers to the basic energy-optimization method that does not include/ consider environmental parameters other than the temperature and humidity. $E_{cost}$ refers to the associated cost in controlling the energy, whereas $E_{temp}$ and $E_{humid}$ refer to the energy costs of controlling temperature and humidity, respectively. The former is as follows:(9)$$E_{cost}=E_{temp}+E_{humid}$$

Equation (Equation 10) refers to the proposed energy-optimization method that considers weather conditions along with the temperature and humidity factors. The proposal introduces the use of weight factors for each optimization parameter, which is based on the data analysis performed with the features. $W_{temp}$, $W_{humid}$, $W_{airpressure}$, $W_{dewpoint}$, and $W_{windspeed}$ refer to the weights associated with temperature, humidity, air pressure, dew point, and wind speed, respectively. The goal of this formula is to provide a minimal cost for controlling the appliance energy. Based on this cost, we assume that these metrics reduce the energy of an appliance. The optimization formula in Equation (Equation 10) is represented as follows:(10)$$\begin{array}{cc} E_{cost} & =w_{temp}*E_{temp}+w_{humid}*E_{humid}\\ \; & +(w_{airpressure}+w_{dewpoint}+w_{windspeed}) \end{array}$$

In Algorithm 1, the current state of the environment is fetched. The number of particles in a swarm is the number of iterations, and the optimal solution is found accordingly. This approach considers the minimization function explained in Equation (Equation 10). The user-desired settings are also considered in optimization, which are temperature and humidity ranges. Based on this information, the updated cost calculated and finds the minimum value, as the goal of this work is to minimize the cost spent on controlling cost. The total $E_{cost}$ is compared with previous work, and the proposed Algorithm 1 shows better performance in terms of optimizing the energy cost of the provided environment.
**Algorithm 1:** Energy-optimization algorithm. $State_{current}$←getCurrentState() **for** *i* in $State_{current}$ **do**    $C_{best}$←PSO(      $State_{current_{temperature}}$, $State_{current_{humidity}}$, $State_{current_{airpressure}},$      $State_{current_{dew-point}}$, $State_{current_{wind-speed}}$, $State_{num_{swarm-particles}},$      $User_{settings}$, MimizationFunction from Equation (Equation 10)     )    **for** *j* in $num_{swarm-particles}$ **do**      Apply Cost Function      **if** $C_{j.curr}$≤$C_{best}$ **then**         Select and update $C_{best}$ for particle *j*         Update $V_{best}$ Velocity         Update $P_{best}$ Position      **end if**      Update Solutions Cost Table    **end for**    Select Best Solution from Cost Table    Update $E_{total-cost}$ **end for**

Overall, the proactiveness and optimization shown in Section 3.2 enable the proactive control, optimization, and scheduling of energy-optimization systems for smart homes.

## 4. Experiment Results

In this section, the process of setting up the testing environment is explained. The details about dataset and its preprocessing are explained. At the end, the evaluations are made in terms of RMSE for the predictions and actual energy costs when controlling appliances for energy optimization.

### 4.1. Test Environment

The environment used for experimentation is shown in Table 4. All experiments were performed on an Ubuntu operating system. Python was used for designing the appliance energy forecast model and the simulation of optimization. Other packages and libraries were used, including Conda, Keras, TensorFlow, Sci-Kit learn, Fuzzy-Module, Matplotlib, Numpy, and Pandas.

### 4.2. Dataset

The dataset [11] shown in Table 5 was used in the study [39], and it contains 19,735 records and 29 columns. The data are categorized as indoor and outdoor data. Of these categories, some of which are temperature and humidity, represented by *T* and *H*, for rooms inside a home and nearby a station. The dataset also contains some information about the light, visibility, air pressure, wind speed, dew point, etc., from the nearest weather station at Chievres Airport, Belgium. The dataset’s accuracy [40] shows that the data are valid and ready to use for analysis and contain useful insights regarding monthly, weekly, daily, and hourly energy usage patterns.

The second dataset [12] shown in Table 6 also contains data that include four seasons, i.e., 12 months data. It contains the temperature and humidity values over 10 min intervals, which are associated with the energy consumption of the house. Evaluation of optimization was performed using this dataset, and it resulted in better performance in terms of energy saving. The experiments performed over this data were based on seasonal, monthly, and daily energy usage patterns.

The preceding dataset was used for experimentation for the following reasons:The dataset has been used in previous studies in our laboratory. It was used to provide optimal control parameters with the use of PSO based on temperature and humidity, and user-desired environmental conditions in the house.The dataset contains four seasons’ data representing different weather conditions. It also contains information other than temperature and humidity, such as air pressure, dew point, and wind speed. This type of data provides the ability to realize the performance better.

### 4.3. Evaluation

This section provides insights about the evaluation of the appliance-energy-prediction model and energy-optimization technique.

#### 4.3.1. Preprocessing

This section describes the process of preprocessing the dataset. This is required to enhance the performances of both prediction and optimization models.

The correlation formula is shown in Equation (Equation 11), which results in the correlation analysis shown in Figure 5. In this equation, $X_{i}$ represents a feature to be compared with $Y_{i}$ from the dataset, whereas the μ and ν represent their respective means.
(11)$$Cov\left(X,Y\right)=\frac{\sum \left(X_{i}-\mu \right)\left(Y_{j}-\nu \right)}{n}$$

Based on the analysis of correlation graph shown in Figure 5, we concluded that some of the features needed to be dropped. Thus, several room temperature and humidity features were dropped: outdoor temperature and light, and both random-variable features due to their low impact on the appliance energy and due to the abundance of zeros values for them. The columns that were dropped overall were lights, $RH_{4}$, $RH_{5}$, $T_{6}$, $T_{9}$, Visibility, Tdewpoint, $rv_{1}$, and $rv_{2}$. The column lights was dropped due to the high number of zeros in it which could impact a model’s performance. $T_{6}$ was removed as it was redundant and almost contained the same values as $T_{o}ut$. The other columns having lower scores than 0.02 were removed in this work.

For experimentation, we utilized two popular scaling techniques, i.e., minmax scaler and standard scaler. From the results shown in Table 7, we found out that minmax-scaler helps results are better than the ones using the standard scaler. The minimum and maximum values set for scaling were −1 and 1, respectively.

Utilizing the filtered feature set, the importance of each feature was calculated, which provided more insights on the need for associating their weights to be used in training process, initially. The feature importance values are shown in Figure 6. These are also termed as coefficients in this manuscript, which are considered to be the weights associated with each feature used in the proposed system. The term coefficients is often replaced with the level of importance in this manuscript to show the weight *w* of each feature separately, and to also show its level of impact on the appliance energy.

Overall, based on preprocessing results, the training of the prediction model and simulation of optimization model can be evaluated. Due to the addition of weight factors, the effects on the models’ performances are evaluated in the following sections.

#### 4.3.2. Prediction

In this section, the appliance energy forecast model is evaluated by its RMSE score over the time. The performance of this model was tested using the dataset shown in Table 5.

The RMSEs when using the other approaches [41] are compared to our RMSE results. The RMSE values shown in Table 7 are for three different months, i.e., February, March, and April. It can be seen that the evaluations vary for each day of the month, referring to the fact that each day of the month has different requirements. This highlights the need for a model that can deal with the patterns of data from different months, days, weekdays, and weekends.

The results for the energy forecast model evaluation are visualized in Figure 7, which reflect the performances of the predictions. The orange and blue colors refer to the forecast and actual appliance energy. Figure 7a–f reflect the prediction comparisons and RMSE scores for the months of February, March, and April.

In the evaluations shown in Table 7, it can be seen that the RMSE is low, as shown in the predictions in Figure 7. We found that 10 splits gives the best performance if MinMax scaling is used with minimum and maximum values of 1 and −1, respectively. Performance was analyzed and evaluated for all months, and the results for the months of February, March, and April are shown in Table 7.

#### 4.3.3. Optimization

In this section, the energy optimization model is evaluated by its energy cost when controlling the appliances in consideration of user-desired room settings. Simulations of the proposed hybrid optimization approach, i.e., PSO-GWO, and PSO-based optimization reliant on temperature and humidity metrics [30], showed that the proposed approach saves more energy cost when controlling the appliances in a smart home.

For evaluation, we used two optimization techniques specified in Table 8. The PSO-based method, our previous work, makes use of temperature and humidity metrics as per Equation (Equation 9). The proposed optimization model, the PSO-GWO approach, makes use of additional metrics of the model as well, i.e., temperature, humidity, air pressure, dew point, and wind speed; and the formulation can be seen in Equation (Equation 10). In addition, weight-factors based on analysis are applied in the proposed optimization model.

The energy savings using the previous approach [31] that uses PSO as the model and two factors, temperature and humidity, are compared with the proposed optimization technique’s results (it makes the use of other weather factors in its cost function). The model configurations are shown in Table 8.

The criteria for the user-desired temperature and humidity were set to 25 and between 53 and 56, which are shown in the form of Table 9. It can be seen that if the the temperature is cool or hot, we have defined the criteria for keeping the user-desired temperature by taking appropriate actions. The same hold for the humidity level.

The metrics, temperature and humidity, are the user-desired metrics used in this work. Other metrics, i.e., air pressure, dew point, and wind speed, have shown effects in Table 10 due to their inclusion in the cost function of the optimization procedure formulated in Equation (Equation 10). The costs associated with controlling the appliances according to the user-desired settings are compared between the proposed PSO-GWO- and PSO-based optimizations [31]. They differ in their cost functions, depicted in Equations (Equation 9) and (Equation 10), respectively. Due to the fluctuation of energy consumption in seasons, the performance of the optimization model might be affected. Therefore, season-wise performance comparison was tabulated to prove the credibility of the proposed optimization technique, due to the use of weather factors and their weights other than temperature and humidity.

It is also a known fact energy consumption fluctuates among the months, which might affect the performance of the optimization model. Even in different months, the optimization performs well, hence proving the credibility of the weight of each factor. Due to its better performance, the smart home management systems are allowed to configure the optimal control for as long as it is desired.

The total number of saved kW units accumulated for all seasons was 17,094.95, in comparison to the PSO-based approach without the outdoor weather metrics. Table 10 shows that the proposed optimization technique works quite well in the different seasons as well.

In Table 11, a month-wise comparison is provided for the evaluation of the proposed optimization technique, i.e., the PSO-GWO approach. It can be seen in comparison of energy cost in controlling the appliances, between the proposed optimization technique and previous approach [31], that the proposed optimization technique provides better results. It is also a known fact the energy consumption fluctuates by month, which might affect the performance of the optimization model. Even in different months, the optimization performs well, proving the credibility of the weight of each factor. Due to its better performance, the smart home management systems are allowed to configure the optimal control for as long as it is desired.

The proposed optimization model was applied to the second dataset [12]. The relevant results are shown in Table 12, which reflect the optimization in the form of minimizing the cost associated with controlling the appliances according to the user-desired environmental settings of the room.

The total number of saved kW units accumulated for the period of 12 months was 17,091.20, in comparison to the PSO-based approach without the outdoor weather metrics. Table 11 shows that the proposed optimization technique performs better for all months as well.

In Table 13, a comparison of the energy cost by increasing the number of months is provided. It can be seen that the energy cost of the proposed optimization technique, i.e., PSO-GWO, is very low as compared to that of the previous approach [31]. It can also be seen that an increase in the number of months does not affect the performance, which allows the stakeholder to configure the optimal control for as long as desired.

Same results for the dataset [12] can be visualized in Table 14. It is evident that the energy cost optimization is better when the proposed methodology is used as it results in more energy controlling cost savings.

The experiments have shown that the data requirements change by season, month, day, weekday or weekend status, and time of day. This highlights the need for a model that can first identify the factors, and then apply the time-series forecasting of the model. This can be validated by the use of different datasets representing variances over different regions. In this work, we utilized two different datasets, i.e., Chievres Airport, Belgium [11], and Cheonan, South Korea [12]. This definitely enhanced the performance of prediction and optimization in terms of the RMSEs of accuracy and energy-cost saving when controlling the appliances. In addition to this, the GWO- and PSO-based optimizations were formulated together, which can be utilized and applied to the appliance energy forecasts. The experimental results reflect the fact that each weather factor has an important role to play in the prediction and optimization of energy because of the geographical, seasonal, and day-type (weekend, weekday) variations, among others. Based on the results, it is concluded that the inclusion of air pressure, dew point, and wind speed enhance the performance and result in reducing the energy costs involved in controlling the smart home appliances.

## 5. Discussion

In this section, the findings of the current work are discussed. Current energy systems have the limitation of performing with different levels of efficacy depending on the season or month. This is due to the fact that they do not account for the level of impact features related to the time of year have on the energy. Analysis and supporting results from Section 4 showed that our energy-optimization scheme works rather well in all of the seasons and months.

The energy-saving results shown in Table 10, Table 11 and Table 13 also support the idea of considering weight coefficients in existing optimization techniques to enhance their performances in certain seasons and months as well. The values that we used in this work for the dataset are shown in Table 15.

The primary findings of this work are that the inclusion of properly analyzed weight coefficients could enhance optimization techniques and result in energy savings. They also play an important role in the prediction of appliance energy use.

The secondary focus of this work was to consider the level of impact of a specific feature on the energy consumption. For this purpose, the weight coefficients are included in both the energy prediction and optimization models.

From physical point of view, energy savings could be explained as such: if the appliances’ optimal control parameters are not frequently updated, then their energy efficiency is significantly reduced. This wasted energy could be saved for that specific dataset/area where the appliances were unnecessarily turned on.

It can also be deduced from the seasonal and month-wise comparison results that the weight factors play an important role in the optimization problem. In this work, we primarily focused on finding the best weight factors for the region used in this study. It can be confidently said that if the same technique used to estimate the weight factors is applied carefully for another dataset representing different regions with their own weather conditions, the optimization model will perform the same. This is due to the use of weight factors which reflect different regional weather conditions.

## 6. Conclusions

From the previous results, we can conclude that the proposed system can enable proactiveness and efficiency in current energy-optimization systems. It can be stated that this enhancement is provided by the inclusion of weight coefficients that refer to the levels of impact of weather metrics on the appliance energy. This not only enhances the performance of the optimization technique but also that of the prediction technique. with the use of properly estimated weight factors, this optimization approach could be applied to multi-regional datasets. The second interesting point of this work is that it considered the level of impact of each feature, which again refers to the fact that this optimization has a broader scope in terms of its application by using a variety of features.

Further enhancements to the optimization module by optimizing, scheduling, and controlling appliance energy based on season, month, day, week, weekend, and time using GWO-PSO-based optimization showed that the proposed system can provide optimal control parameters through the use of weather metrics other than just temperature and humidity. This addition broadens the scope of both prediction and optimization models and enhances them to perform better in various seasons, months, day-types.

## Figures and Tables

**Figure 1 sensors-23-03640-f001:**
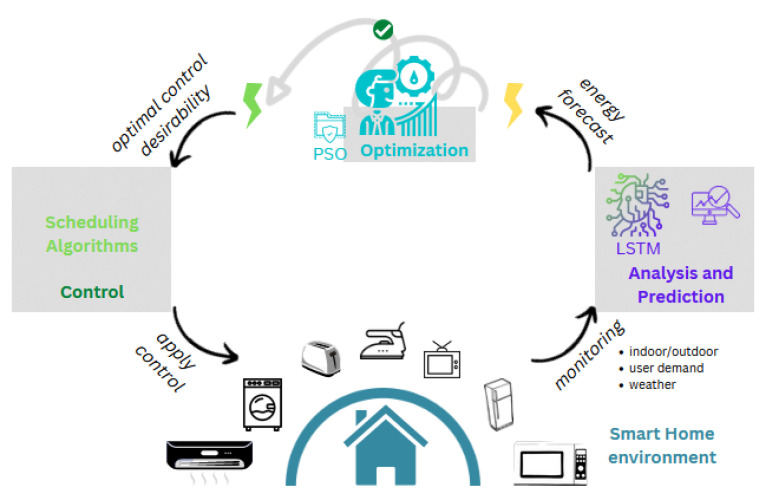
Complex Problem-Solving (CPS) environment: analysis, prediction, optimization, scheduling, and control.

**Figure 2 sensors-23-03640-f002:**
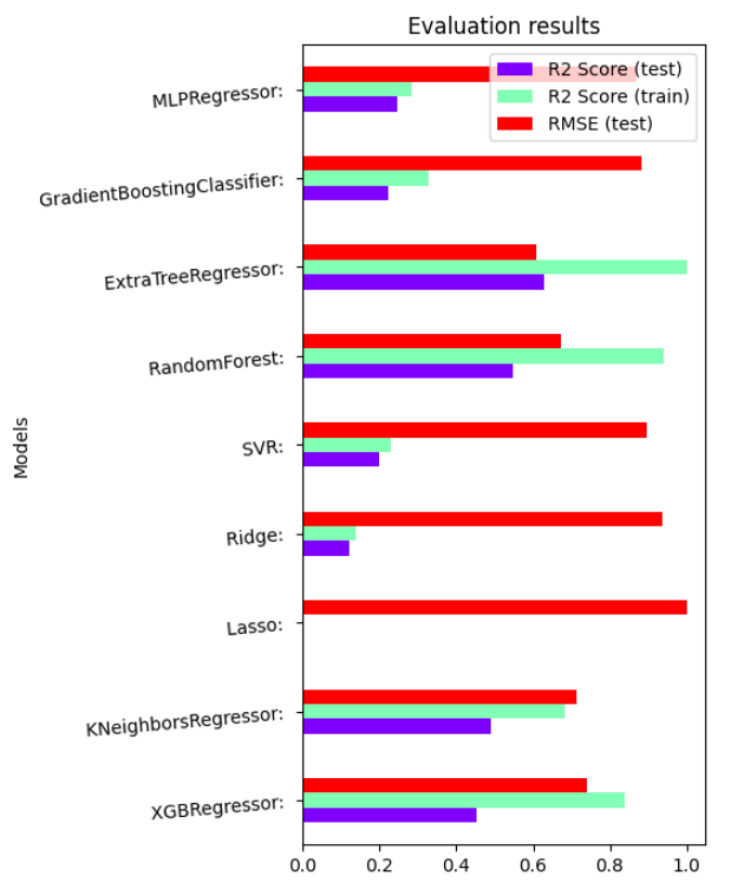
Root Mean Square Error (RMSE) and R2 scores of multiple models.

**Figure 3 sensors-23-03640-f003:**
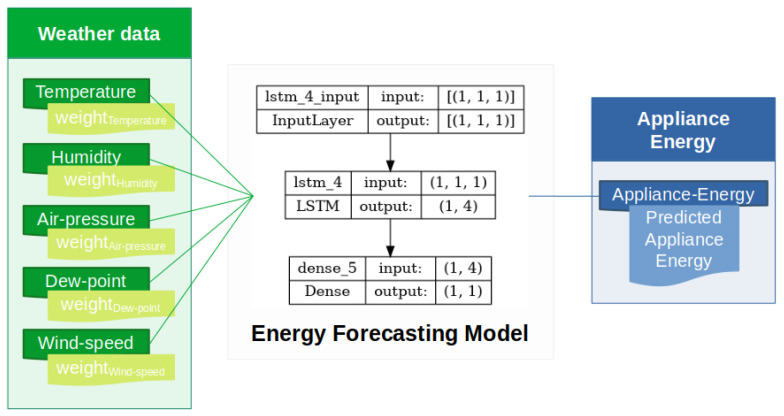
Proposed appliance energy prediction system.

**Figure 4 sensors-23-03640-f004:**
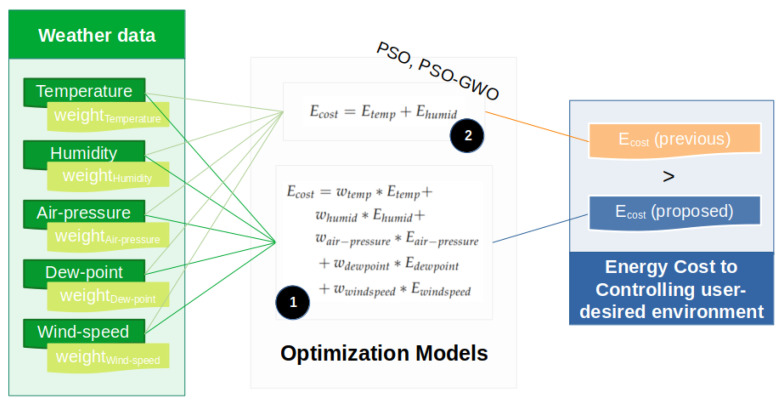
Proposed energy-optimization system.

**Figure 5 sensors-23-03640-f005:**
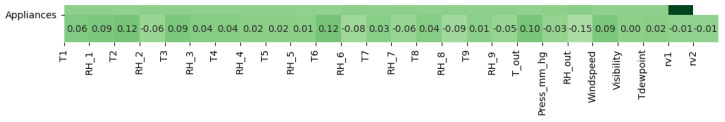
Feature covariance(s).

**Figure 6 sensors-23-03640-f006:**
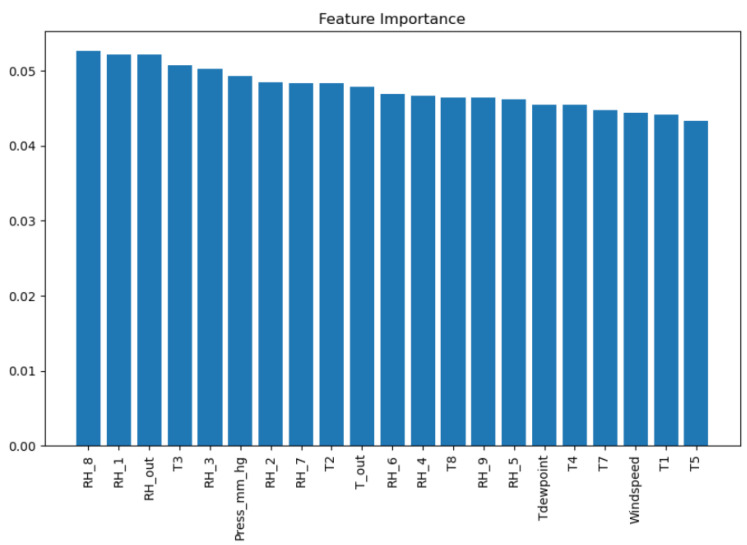
Filtered features’ levels of importance.

**Figure 7 sensors-23-03640-f007:**
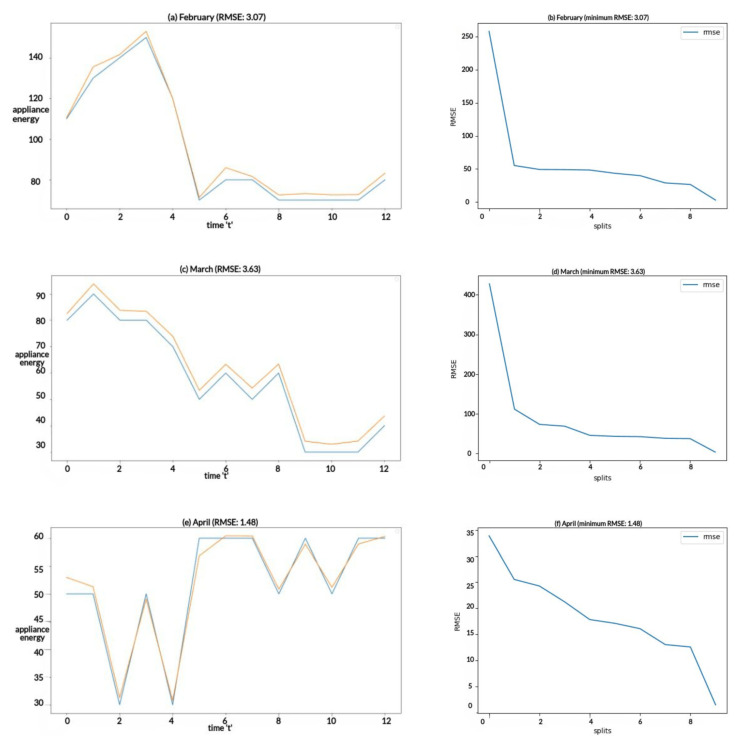
Appliance energy forecasts and RMSE score of the model.

**Table 1 sensors-23-03640-t001:** Data and variables used in scaling.

Symbol	Description
$X^{\prime }$	Scaled dataset, after applying the MinMax transformation
*X*	Actual data, i.e., represented in the form of a vector
*x*	it is an element from the vector *X* representing a feature
$x^{\prime }$	scaled value of element *x* taken from the vector *X* representing a feature
$x_{min}$	it is the smallest element of vector *X*
$x_{max}$	it is the largest element of vector *X*
max	minimum value set for scaling the data, i.e., −1
max	maximum value set for scaling the data, i.e., 1

**Table 2 sensors-23-03640-t002:** Data and variables used in prediction.

Symbol	Description
*t*	represents the time-step of LSTM model training
σ	sigmoid activation function, output between 0 and 1
ϕ	tangent hyperbolic function, output between −1 and 1
×	symbol representing cross product operation
+	symbol representing sum operation
$x_{t}$	input at time-step *t* from a vector *X*
$h_{t-1}$	previous hidden state of an LSTM unit
$i_{t}$	input gate of an LSTM unit
$W_{xi}$	weight of $x_{t}$ used in the input gate $i_{t}$
$W_{hi}$	weight of $h_{t-1}$ used in the input gate $i_{t}$
$b_{i}$	bias for the input gate $i_{t}$ used in an LSTM unit
$f_{t}$	forget gate of an LSTM unit
$W_{xf}$	weight of $x_{t}$ used in the forget gate $f_{t}$
$W_{hf}$	weight of $h_{t-1}$ used in the forget gate $f_{t}$
$b_{f}$	bias for the forget gate $f_{t}$ used in an LSTM unit
$o_{t}$	output gate of an LSTM unit
$W_{xo}$	weight of $x_{t}$ used in the output gate $o_{t}$
$W_{ho}$	weight of $h_{t-1}$ used in the output gate $o_{t}$
$b_{o}$	bias for the output gate $o_{t}$ used in an LSTM unit
$W_{xc}$	weight of $x_{t}$ used in the modified input $\tilde{c_{t}}$
$W_{hc}$	weight of $h_{t-1}$ used in the modified input $\tilde{c_{t}}$
$b_{c}$	bias used for modifying the input $\tilde{c_{t}}$
$\tilde{c_{t}}$	modified input of current LSTM unit
$c_{t-1}$	memory cell of the previous LSTM unit
$c_{t}$	memory cell of an LSTM unit
$h_{t}$	hidden state of current LSTM unit

**Table 3 sensors-23-03640-t003:** Data and variables used in optimization.

Symbol	Description
$E_{cost}$	total cost in controlling the energy
*m*	a metric denoting either temperature, humidity, air-
	-pressure, or wind speed.
$E_{metric_{m}}$	cost in controlling the energy associated with metric *m*
$w_{metric_{m}}$	weight associated with metric *m*
$\hat{t}$	time variable in GWO optimizer
α	alpha, best-fit solution for the optimizer at time-step $\hat{t}$
β	beta, second best solution for the optimizer at time-step $\hat{t}$
γ	gamma, third best solution for the optimizer at time-step $\hat{t}$
$\vec{X}$	mean value of top 3 fits (α, β, and γ) Equation (Equation 8)

**Table 4 sensors-23-03640-t004:** System environment and software packages.

Software	Version	Software	Version
Ubuntu	22	Python	>3.7.x
Conda	22.9.0	Keras	TensorFlow2
TensorFlow	2	Sci-Kit learn	1.2.0
Fuzzy-Module	1.2.2	Matplotlib	3.5
Numpy	1.17	Pandas	1.5.x

**Table 5 sensors-23-03640-t005:** Appliance energy dataset reprinted/adapted with permission from Ref. [11]. 2023, Kaggle-GoKagglers.

Appliance Energy	$T_{1}$	$H_{1}$	$T_{i}$	$H_{i}$	*…*	Air -Pressure	Wind -Speed	Dew -Point	Visibility
60	19.9	47.6	19.2	44.8	*…*	733.5	7.0	63.0	5.3
60	19.9	46.7	19.2	44.7	*…*	733.6	6.7	59.2	5.2
50	19.9	46.3	19.2	44.6	*…*	733.7	6.3	55.3	5.1
*…*	*…*	*…*	*…*	*…*	*…*	*…*	*…*	*…*	*…*
50	19.9	46.1	19.2	44.6	*…*	733.8	6.0	51.5	5.0
60	19.9	46.3	19.2	44.5	*…*	733.9	5.7	47.7	4.9
60	19.9	46.4	19.2	44.6	*…*	734.1	5.4	42.8	4.8

**Table 6 sensors-23-03640-t006:** Appliance energy dataset (South Korea) [12].

Dataset Features	
time	timestamp of the record.
temp	temperature of the room at a given time.
sky	visibility of the sky at time.
ws	wind speed of the region at time.
wd	dew point in the air at time.
wdEn	wind direction at time.
reh	humidity of the room at time.
appl	appliance energy consumed at time.

**Table 7 sensors-23-03640-t007:** RMSE comparison.

Dataset	Month	Day	Scaling	Model	Split	RMSE
		10			10	3.07
[11]	February	15	MinMax (−1, 1)	LSTM	10	5.9
		5			10	6.33
		10			10	3.63
[11]	March	27	MinMax (−1, 1)	LSTM	10	3.64
		13			10	3.78
		2			10	1.48
[11]	April	1	MinMax (−1, 1)	LSTM	10	3.09
		7			10	3.66
		28			10	5.96
[12]	February	16			10	6.57
		27			10	7.28
		4			10	5.15
[12]	March	29	MinMax (−1, 1)	LSTM	10	5.85
		31			10	6.84

**Table 8 sensors-23-03640-t008:** Different model configurations used for comparison.

Optimization Model	Metrics	Use of Weight-Factors?	Particles
PSO [31]	temperature and humidity	No	15
PSO-GWO (proposed)	temperature, humidity, air-pressure, dew-point, and wind-speed	Yes	15

**Table 9 sensors-23-03640-t009:** Criteria for controlling the temperature and humidity.

Metric	Criteria	Status	Action
temperature	16∼25	Cool	Heat the place
	25	Normal	None
	25∼30	Hot	Cool the place
humidity	40∼53	Less Humid	Increase the Humidity level
	53∼56	Normal	None
	56∼65	Too Humid	Lower the Humidity level

**Table 10 sensors-23-03640-t010:** Season-wise energy-cost comparison.

Season	Months	Simulations	Appliance Control		
			Energy Cost		Saved Energy
			Previous	Proposed	Cost in kW
Spring-2014	May	1744	1927.03	522.7	1404.33
Summer-2014	Jun–Aug	32,208	5621.96	1547.59	4074.37
Autumn-2014	Sept–Nov	32,184	5927.82	1547.62	4380.20
Winter-2014–15	Dec–Feb	32,160	6023.88	1538.46	4485.42
Spring-2015	Mar–Apr	21,464	3778.16	1027.53	2750.63
All seasons Energy Cost Saved (season-wise):					17,094.95

**Table 11 sensors-23-03640-t011:** Month-wise energy-cost comparison for dataset [39].

Year	Month	Simulations	Appliance Control		
			Energy Cost		Saved Energy
			Previous	Proposed	Cost in kW
2014	May	744	1926.81	522.7	1404.14
2014	June	720	1863.27	506.04	1357.23
2014	July	744	1857.15	519.77	1337.38
2014	August	744	1898.68	521.85	1376.83
2014	September	720	1916.56	508.49	1408.07
2014	October	744	2007.36	526.55	1480.81
2014	November	720	2002.67	512.57	1490.10
2014	December	744	2096.59	531.0	1565.59
2015	January	744	2097.26	531.04	1566.22
2015	February	672	1830.15	476.41	1353.74
2015	March	744	1949.24	523.65	1425.59
2015	April	720	1829.43	503.9	1325.53
Annual Energy Cost Saved (month-wise):					17,091.20

**Table 12 sensors-23-03640-t012:** Month-wise energy cost-comparison for dataset [12]/Table 6.

Year	Month	Simulations	Appliance Control		
			Energy Cost		Saved Energy
			Previous	Proposed	Cost in kW
2023	February	−−	71.58	19.04	52.54
2023	March	−−	318.96	85.83	233.13
Annual Energy Cost Saved (month-wise):					285.67

**Table 13 sensors-23-03640-t013:** Energy-cost comparison when increasing the number of months.

Start	End	Months	Simulations	Appliance Control	
				Energy Cost	
				Previous	Proposed
2014 May	2014 May.	1	744	1926.83	522.69
2014 May	2014 Jun.	2	1464	3790.77	1028.71
2014 May	2014 Jul.	3	2208	5647.17	1548.45
2014 May	2014 Aug.	4	2952	7549.21	2070.26
2014 May	2014 Sep.	5	3672	9464.63	2578.8
2014 May	2014 Oct.	6	4416	11,474.7	3105.31
2014 May	2014 Nov.	7	5136	13,476.65	3617.95
2014 May	2014 Dec.	8	5880	15,574.62	4148.77
2014 May	2015 Jan.	9	6624	17,669.89	4679.95
2014 May	2015 Feb.	10	7296	19,503.27	5156.34
2014 May	2015 Mar.	11	8040	21,449.89	5679.94
2014 May	2015 Apr.	12	8760	23,456.78	6183.92

**Table 14 sensors-23-03640-t014:** Energy-cost comparison when increasing the number of months for dataset [12].

Start	End	Months	Simulations	Appliance Control	
				Energy Cost	
				Previous	Proposed
2023 Feb	2023 Feb.	1	27	71.58	19.04
2023 Feb	2023 Jun.	2	122	318.96	85.83

**Table 15 sensors-23-03640-t015:** Coefficients and weight values.

Coefficients	Values	Coefficients	Values
$RH_{1}$	0.0515	$T_{1}$	0.0430
$RH_{2}$	0.0484	$T_{2}$	0.0478
$RH_{3}$	0.0500	$T_{3}$	0.0501
$RH_{4}$	0.0469	$T_{4}$	0.0454
$RH_{5}$	0.0463	$T_{5}$	0.0420
$RH_{6}$	0.0470		
$RH_{7}$	0.0482	$T_{7}$	0.0450
$RH_{8}$	0.0520	$T_{8}$	0.0467
$RH_{9}$	0.0465		
$RH_{out}$	0.0512	$T_{out}$	0.0478
$T_{dewpoint}$	0.0455	*Wind-speed*	0.0440
*Air-pressure*	0.4900		

## Data Availability

Dataset 1: https://www.kaggle.com/datasets/loveall/appliances-energy-prediction (accessed on 26 March 2023). Dataset 2: https://doi.org/10.21227/9m0k-cm61 (accessed on 26 March 2023).

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
