# Peer review of "Energy Prediction and Optimization for Smart Homes with Weather Metric-Weight Coefficients"

_sensors, 2023, doi:10.3390/s23073640_

Round 1

Reviewer 1 Report

The paper is devoted to actual topic of energy consumption prediction based on machine learning approach. However, to estimate the paper under review a number of amendments and clarifications are required. The paper had to be sufficiently reconsidered and improved.

1. The authors propose update of Ecost evaluation by introducing new weather parameters and coefficients (wi). The meaning of this parameters had to be explained. The paper does not describe method of calculation and their specific values.

2. How the training of model was performed? RMSE for which parameters has been estimated?

3. Tables 6-8 show more than three-fold reducing of Energy cost. How it can be explained from physical point of view?

4. Quality of Fig. 6 is not sufficient. There are not many required attributes (axis labels, legends, etc.).

5. Line 412 comprises the phrase “importance of each feature is calculated”. What does importance mean? Please, provide the corresponding formulas.

6. Fig. 4 is unclear. How the feature covariance calculated? What do parameters T1, RH1, T2, RH2, etc. mean? Why the covariance between Appliances and Appliances is so small (-0.01)?

7. Section 4.2 describes three different datasets, which one is used?

8. Minimum of eq. 1 is zero. However, authors in line 285 say that the minimal value is -1.

9. How the specific configuration of the neural network is selected? Why do use LSTM? Please, describe the output of the proposed neural network, as well as its input data.

Author Response

I am very thankful to the reviewer for comments. The comments are addressable and have been updated according to my knowledge. Point-by-point responses are as follows:

  1. The name of the method was added right after eq. 1. As per the review, more explanation about the calculation is added in section 3.1, i.e., Proposed system -> Pre-processing. To get the best hyperparameter configurations, we applied multiple models, e.g., MPL Regressor, Gradient Boost Classifier, Extra Tree Regressor, Random Forest, SVR, Ridge, Lasso, K-Neighbors Regressor, XGB Regressor. Among all these models, the Extra Tree Regressor outputs the best results with the least RMSE score and maximum R2 Score. Based on these results, we applied Boruta algorithm to get the weight coefficients which we call importances as well. In this section, fig. 2 (newly added) depicts one of the steps involved in getting the weight co-efficients.
  2. Training of model was performed based on the configurations shown in fig. 3 (previous version fig. 2). The training details of this model are added in the explanation as well. We trained the model with 50 epochs with the learning rate of 0.01. RMSE for appliance energy has been estimated. As per the reviewer's comment, its explanation has been added to the paper.
  3. This explanation is provided in a newly created section, i.e., Discussions. Experiments were performed in the form of a simulation. From physical point of view, it could be explained as that if the appliances optimal control parameters are not frequently updated, then the energy is significantly reduced. Secondly, this much energy could be saved for that specific area where the appliances were unnecessarily turned on. This could be due to several reasons.
  4. As per the reviewer's comment, quality of fig. 6 has been improved. Axis, x and y are labled now.
  5. In this article, the word importance is associated with the level-of-impact the feature has to appliance energy consumption. After revising the whole manuscript, this issue has been resolved and proper word for it has been replaced where confusion. The corresponding algorithm and its explanation has been provided when addressing reviewer's comment/point # 1
  6. Covariance of calculated based on a formula, i.e., Cov(X, Y) = Σ(Xi-µ)(Yj-v) / n, where X is the set of input features such as T1-T9, RH_1-RH-9, Pressure_mm_hg, Windspeed, dewpoint. While Y represents the target fetaure, i.e., appliance energy. T1 refers to temperature of room 1, RH_1 refers to humidity of room 1. The covriance relationship and its explanation are being added to the revised submission. This part has been added in the explanation of equation 11.
  7. The dataset shown in table 5 (previous manuscript table 3) was used for prediction of appliances, while the second data-set refered from two articles is the same dataset [11, 12], which was used for optimization of appliances.
  8. We meant to say that this one of the standard approaches to scale the dataset. We set the min value to -1, and max as 1. The equation 1 always scales the input/feature to a value between -1 and 1. The explanation has been updated just in case to remove ambiguity.
  9. From previous works, LSTM model was evaluated to result in better results. This being the reason, I selected LSTM for experimentation, and added the weight factors to the training. All feature such as temperature, humidity, windspeed, air-pressure, dewpoint were selected as inputs while the appliance energy was selected as output. Rest of the columns were dropped as explained in preprocessing section. The explanation describing the reason of removing several columns has been updated as well. The model output is also described when explaining fig. 2. Another table like the table 5 (current manuscript) is being added which will show the prediction model details or a figure will be added to describe better.

Reviewer 2 Report

See attached review comments.

Author Response

I am very thankful to the reviewer for comments. The comments are addressable and have been updated according to my knowledge. Point-by-point responses are as follows:

  1. List of Abbreviations have been added before the Acknowledgment section. This was required to enhance the paper and helped organizing the manuscript content.
  2. Grammatical mistakes have been corrected. many other mistakes have been corrected after revision.
  3. Rearranged Table 1, under a separate subsections an split them in different tables. Added the tables under each respective subsection. Name of each table is captioned with respective association.
  4. All table and figure captions have been aligned below.
  5. Point 5 reviews are as follows:
    - The energy savings shown in table 8,9,10 results also support the ideology of considering weight coefficients in existing optimization techniques to enhance its performance over different seasons and months as well.
    - The primary findings of this work are that the inclusion of properly analyzed weight coefficients could enhance the optimization techniques, and result in energy savings. It also plays an important role in the prediction of appliance energy.
    - The secondary focus of this work is to consider the level of impact of a specific feature over the energy consumption. For this purpose, the weight coefficients are included in both the energy prediction and optimization models.
    - It can also be deducted from the seasonal and month-wise comparison results that the weight factors play an important role in the optimization problem. In this work, we have primarily focused on finding the bets weight factors for the region used in this region. It can be confidently said that if the same technique to estimate the weight factors is applied carefully for another data-set representing different regions with its own weather conditions, the optimization model will perform the same. This is due to the fact of using weight factors which reflect different regional weather conditions.
  6. Point 6 reviews are as follows:
    - From the previous results, we can conclude that the proposed system can enable proactiveness along with efficiency in current energy optimization systems. It can be stated that this enhancement is due to the inclusion of weight coefficients that refer to the level of impact of a weather metric over the appliance energy. This not only enhances the performance of optimization technique but also the prediction technique. By the use of properly estimated weight factors, this optimization approach could be applied to multi-regional data-sets. The second interesting point of this work is that it considers the level of impact of each feature, which again refers to the fact that this optimization has a broader scope in terms of its application with a variety of features.
    - Further enhancements to the optimization module by optimizing, scheduling, and controlling appliance energy based on season, month, day, week, weekend, and time using hybrid GWO-PSO-based optimization shows that the proposed system can provide optimal control parameters through the use of weather metrics other than just temperature and humidity. This addition broadens the scope of both prediction and optimization models and enhances them to perform better on different seasons, months, day-types.

Round 2

Reviewer 1 Report

1. The article introduces the coefficients w in the formula (10). What values can these coefficients take? What will mean, for example, a coefficient of 0.5? Why are the specific values of these coefficients obtained for the considered data set not given in the article?

2. The parameters Etemp and Ehumid have a clear meaning: the cost of energy to control temperature and humidity. What is the meaning of parameters Eairpressure, Edewpoint and Ewindspeed?

3. What does cost function optimization (line 494) mean? What optimization criterion was used? What was the cost function (10) compared to during optimization?

4. The triple energy savings is explained by new appliance control strategy? Can you describe it in more detail? With the dataset in question, without additional data, how can the claimed savings be justified?

5. Equation (1), as it is presented, does not allow normalizing to the interval from -1 to 1. If you disagree, please give a specific example of the vector X, in which negative values will appear after applying equation (1). From my point of view, the formula for normalizing to a segment from -1 to 1 should have the following form: Tminmax=-1+2(x-xmin)/(xmax-xmin).

6. Please explain why the appliance-to-appliance covariance has a negative value of "-0.01" (Figure 5). If formula (11) is used, at least a positive number should be obtained.

Cov(Y,Y)=Σ(Yj-ν)(Yj-ν)/n= Σ(Yj-ν)2/n

7. Fig. 7 needs to be improved, at least in terms of font size and axis label placement

8. What does the term importance used in Fig. 6 mean?

9. What specific features were dropped out after the analysis of Fig. 5?

Author Response

  1. The coefficient values have been shared in the section 5, i.e., Discussion. These values are meant to reflect/show the level of impact of that specific feature in the appliance energy consumption. The value of 0.5 means that feature has half impact on the energy consumption of appliance. But we haven't used 0.5 in our work. The values have been in the form of a table in the revised manuscript.
  2. The Edewpoint, Eairpressure, and Ewindspeed were just used as a value of 1 in the equation. We value the comment of reviewer and the reason being, we have removed these parameters from the equation as it was causing confusion and had no meaning. We have updated the equation to use only the weight metrics.
  3. The equation (10) has been explained. It is used to minimize the cost of controlling appliance energy. This cost of controlling energy is minimized (optimized) by the use of optimization model, i.e., PSO-GWO hybrid, which is explained in subsection 3.3, i.e., Optimization. The criteria is based on a fuzzy logic, i.e., if the temperature lies between 16-25 then heating is needed,  if it is above 25 or so cooling is needed. While in the case of humidity, if it lies between the range of 43~53 then there is a need to increase the humidity level a little, or if it is in between 56~56 then there is a need to decrease the humidity level. Each, temperature and humidity control has an associated cost which we use in the cost function/equation. This criterion has been added to the manuscript (table 9) and has been clarified. Thank you for this comment.
  4. The triple energy saving has been now justified by the use of two datasets. One additional experimentation result is also added (by the use of a new dataset taken from South Korea, Cheonan city). The optimization and prediction results have been added to the relevant sections. The results show better performance in both regions, i.e., Belgium and South Korea appliance energy consumption dataset.
  5. Equation 1 was incorrect, and we are thankful to the reviewer for the correction. It has been updated.
  6. The last value, i.e., -0.01, is associated to the metric rv2. It is not associated to appliance. It is as the matplotlib library works. I know that this should not be shown to avoid confusion. Although it is being shown from the matplotlib library side, I have removed by editing the image. For your surity, I have added a image file in the manuscript figures folder, i.e., figures/eval-covariance-full.png.
  7. Fig. 7 has been improved as per the suggestions. 
  8. From fig. 6, the term importance is often interchangeably used in place of weight/coefficient. We have used this term to present that each feature has its own level of impact on the appliance energy. This explanation has been made clear and better explained in the manuscript.
  9. These features lights, RH4, RH5, T6, T9, Visibility, Tdewpoint, rv1, rv2 were dropped due to several reasons by the analysis. lights was removed due to the high number of zeros. T6 due to its redundancy with Tout. Others due to their low covariance scores (lesser than 0.02) were dropped.